# Pulsed Electric Fields to Improve the Use of Non-*Saccharomyces* Starters in Red Wines

**DOI:** 10.3390/foods10071472

**Published:** 2021-06-25

**Authors:** Cristian Vaquero, Iris Loira, Javier Raso, Ignacio Álvarez, Carlota Delso, Antonio Morata

**Affiliations:** 1enotecUPM, Chemistry and Food Technology Department, ETSIAAB, Universidad Politécnica de Madrid, Avenida Puerta de Hierro 2, 28040 Madrid, Spain; c.vaquero@upm.es (C.V.); iris.loira@upm.es (I.L.); 2Tecnología de los Alimentos, Facultad de Veterinaria, Instituto Agroalimentario de Aragón-IA2, Universidad de Zaragoza-CITA, c/Miguel Servet, 177, 50013 Zaragoza, Spain; jraso@unizar.es (J.R.); ialvalan@unizar.es (I.Á.); carlotad@posta.unizar.es (C.D.)

**Keywords:** pulsed electric fields (PEF), colour extraction, vitisin A, *Lachancea thermotolerans*, lactic acid, *Hanseniaspora vineae*, 2-phenylethyl acetate, *Torulaspora delbrueckii*, 3-ethoxy-1-propanol

## Abstract

New nonthermal technologies, including pulsed electric fields (PEF), open a new way to generate more natural foods while respecting their organoleptic qualities. PEF can reduce wild yeasts to improve the implantation of other yeasts and generate more desired metabolites. Two PEF treatments were applied; one with an intensity of 5 kV/cm was applied continuously to the must for further colour extraction, and a second treatment only to the must (without skins) after a 24-hour maceration of 17.5 kV/cm intensity, reducing its wild yeast load by up to 2 log CFU/mL, thus comparing the implantation and fermentation of inoculated non-*Saccharomyces* yeasts. In general, those treated with PEF preserved more total esters and formed more anthocyanins, including vitisin A, due to better implantation of the inoculated yeasts. It should be noted that the yeast *Lachancea thermotolerans* that had received PEF treatment produced four-fold more lactic acid (3.62 ± 0.84 g/L) than the control of the same yeast, and *Hanseniaspora vineae* with PEF produced almost three-fold more 2-phenylethyl acetate than the rest. On the other hand, 3-ethoxy-1-propanol was not observed at the end of the fermentation with a *Torulaspora delbrueckii* (Td) control but in the Td PEF, it was observed (3.17 ± 0.58 mg/L).

## 1. Introduction

Pulsed electric fields (PEF) are a nonthermal technique that causes electroporation of cell membranes by applying very short pulses (µs) of a high-intensity electric field (kV/cm). Irreversible electroporation leads to the formation of permanent conductive channels in the cytoplasmic membrane of the cells resulting in the loss of cell viability, leakage of cytoplasmic compounds and loss of cell turgor [1]. The food industry may take advantage of the effect produced by PEF in microbial cells and cells of animal and plant tissues to improve many different operations. The ability of PEF to inactivate vegetative cells of microorganisms, enhance mass transfer, and modify food structure may contribute to improving the competitiveness of the food industry by improving food quality, reducing energy inputs, and contributing to the bioeconomy strategy for sustainable growth [2].

Wineries can also benefit from PEF in improving several kinds of processes in winemaking. Several studies have demonstrated that electroporation of red grape skin cells before the maceration-fermentation step reduces the duration of maceration and/or improves a wine’s colour and concentration of polyphenolic compounds without impairing its sensorial attributes [3,4,5]. On the other hand, PEF has been proposed as a technique to enhance wine quality by guaranteeing reproducible fermentations and reducing or replacing the use of SO_2_ for wine stabilisation due to the capability of PEF to inactivate microorganisms while preserving physicochemical and sensorial properties [3].

One of the most popular alcoholic beverages in the world [6], wine comes from grape fermentation and has a variety of microbiota on the grape skin that can deteriorate the wine, such as yeasts of the *Dekkera/Brettanomyces* genus and lactic acid bacteria [7,8]. It is controlled with SO_2_, which poses certain health risks [9] and worsens sensory analysis. With this new technology (PEF), it is possible to inactivate unwanted microbiota [10], increase the extraction of phenolic compounds and colour [11], and decrease the SO_2_ dosage [12]. Furthermore, PEF technology can generalise more stable fermentations with selected non-*Saccharomyces* yeasts that provide higher sensory quality such as *Hanseniaspora vineae*, *Torulaspora delbrueckii*, *Lachancea thermotolerans*, *Schizosaccharomyces pombe* and *Metschnikowia pulcherrima* [13,14,15]. The relevance of these non-*Saccharomyces* yeasts has increased in recent years [16]. One of the major problems that some of these yeasts pose with respect to *Saccharomyces cerevisiae* is their weak implantation in addition to their low/medium fermentation power, which makes them poor competitors and which yields a low population from the vineyard [17]. The objective of this work is to evaluate the fermentation effect of different non-*Saccharomyces* yeasts on oenological and sensory characteristics of red wine from Grenache grapes. To obtain this objective, crushed grape was first treated by PEF to obtain a must with a sufficient content of polyphenols after few hours of maceration. Secondly, after removing the grape must, the must was treated again by PEF to reduce the indigenous flora and facilitate the implantation of non-*Saccharomyces* yeasts. 

## 2. Materials and Methods

### 2.1. Grape Sample

For this trial, one hundred kilograms of *Vitis vinifera* var. Grenache grapes were harvested manually in the Campo de Borja appellation, located in the Spanish region of Aragón, at the end of September 2020.

### 2.2. PEF Processing

#### 2.2.1. PEF Unit

A PEF generator EPULSUS PM1-10 (Energy Pulse Systems LDA, Lisbon, Portugal) was used for the application of PEF treatments. This generator, with an output voltage of 10 kV and a maximum current intensity of 200 A, provides monopolar rectangular pulses with a pulse duration between 1 and 200 µS at a maximum frequency up to 200 Hz. Voltage and electric current were measured using a digital oscilloscope (TDS 3021, Tektronix, Wilsonville, OR, USA).

#### 2.2.2. PEF Processing of Grapes

Before PEF processing, grapes were de-stemmed and crushed (Figure 1) (Master E-10 destemmer, Enomundi, Zaragoza, Spain). Grape must (electrical conductivity 1.4 ± 0.1 mS/cm) was pumped by an eccentric screw pump through a co-linear treatment chamber. The co-linear treatment chamber, with an inner diameter of 20 mm, consisted of three stainless steel electrodes with the central electrode connected to high voltage and the outer electrodes connected to the ground. This treatment chamber configuration defines two cylindrical treatment zones with a gap of 20 mm and a diameter of 20 mm. As in a co-linear configuration, the electric field strength is not homogeneously distributed; the electric field strength used to characterise the PEF treatment corresponded to the field strength in the middle position of the central axis of the treatment zones. The flow rate (120 kg/h) provided a residence time of the crushed grape in the treatment zone of 0.38 s. A total treatment time of 1800 µS (45 pulses of 40 µS) was applied at an electric field strength of 5 kV/cm, corresponding to a total specific energy of 63.4 kJ/kg. The temperature of the crushed grape before the treatment was 17 °C, and after the treatments, the temperature was increased at 32 ± 2 °C. Just after processing, the temperature was rapidly decreased to 10 °C using a cooling plate (4 °C) and two lots of 30 kg were distributed in 50 L stainless steel tanks. The tanks were maintained in a cooling chamber (5 °C) for the duration of maceration (24 h) to prevent microbial growth. After 24 h of maceration, Granache must was obtained after pressing the crushed grape with a water press (Enomundi, Zaragoza, Spain) at a maximum pressure of 1 bar.

#### 2.2.3. PEF Processing of Must

Grenache must obtained from grapes treated by PEF and macerated for 24 h at 5 °C (Figure 2) was decontaminated by a PEF treatment of higher intensity (17.5 kV/cm) to facilitate the implantation of non-*Saccharomyces* starters.

A pump (Ismatec, Glattbrugg, Switzerland) was used to pump the must from a reservoir through silicone tubes to a coiled stainless steel tube (Øin 5 mm, Øout 5.5 mm, 230 cm length), which was immersed into a heating bath located before the treatment chamber. The treatment chamber consisted of two parallel stainless steel electrodes (30 mm × 5 mm) with a gap of 4 mm. After the treatment chamber, the temperature of the must was immediately decreased through a similar stainless steel tube immersed into a cooling bath at 4 °C. Grape must (electrical conductivity 1.67 ± 0.1 mS/cm) was pumped through the treatment chamber with a throughput of 10 L/h, providing a residence time of the must in the treatment zone of 0.24 s. A total treatment time of 225 µs (45 pulses of 5 µs) was applied at an electric field strength of 17.5 kV/cm, corresponding to a total specific energy of 115.0 kJ/kg. The temperature of the grape must before the treatment was 25 °C, and after the treatments, the temperature increased to 53 ± 2 °C. After PEF treatment, the temperature of the must decreased in a few seconds to below 15 °C in the heat exchanger located after the treatment chamber. To evaluate the effect of PEF treatment in the extraction of polyphenols, untreated grapes were processed similarly to PEF-treated grapes but with the PEF generator switched off. All the treatments were conducted in duplicate. 

Grenache must obtained from grapes treated by PEF and macerated for 24 h untreated (8 L) and PEF-treated (8 L) was used to evaluate the implantation of non-*Saccharomyces* starters.

The resulting must had a pH of 3.69 ± 0.03 in the PEF-treated must and 3.65 ± 0.02 in the untreated must, and a sugar content of 259 ± 0.1 g/L in the PEF-treated must and 255 ± 0.2 g/L in the untreated must. The 6.84 L of must were distributed into pre-sterilised 250 mL ISO flasks (190 mL per flask) using their own unsealed cap. Fermentations were carried out in triplicate at 19 ± 0.1 °C.

### 2.3. Yeast Used

All yeast ferments were cultured in YPD (yeast extract, peptone, dextrose agar; 10:20:20 g/L) with two sequential steps of 24 h each to homogenise the populations, and then 2% *v/v* was inoculated into the must. This inoculation ratio produced a final population of 6 log CFU/mL. Fermentation lasted for 20 days, and the basic oenological parameters were monitored daily using Fourier-transform infrared spectroscopy (FTIR) and enzymatic analysis. They were also placed in 100 mL sterile vials in triplicate and 60 mL of the (uninoculated) PEF-treated and untreated must was covered with a Müller valve to follow the fermentation.

The following non-*Saccharomyces* and *Saccharomyces cerevisiae* yeast strains were used in the fermentations: the *Hanseniaspora vineae* (Hv205) yeast strain used in this study was isolated by Professor Francisco Carrau (Facultad de Química, Universidad de la República, Montevideo, Uruguay); *Lachancea thermotolerans* (Lt) strain L31 (enotecUPM, ETSIAAB, UPM, Madrid, Spain); *Torulaspora delbrueckii* (Td) strain 291 (Lallemand Inc., Montreal, QC, Canada); *Metschnikowia pulcherrima* (Mp) strain Flavia™MP346 (Lallemand Inc., Montreal, QC, Canada); *Schizosaccharomyces pombe* (Sp) strain 938 (IFI-CSIC, Madrid, Spain); together with the *Saccharomyces cerevisiae* (Sc) strain 7VA (enotecUPM, ETSIAAB, UPM, Madrid, Spain).

### 2.4. Fermentation Trials

A single small-scale fermentation was performed in triplicate using the five non-*Saccharomyces* yeasts plus the *Saccharomyces* yeasts with and without PEF treatment. The procedure was, on one hand, a single fermentation with Sc used as control and sequential fermentations with Lt→Sc, Hv→Sc, Td→Sc, Mp→Sc and Sp→Sc, and on the other hand the same but with the untreated must used to evaluate the inoculum yield and determine the kinetics of the fermentations and their potency.

### 2.5. Yeast Population Counts

Inoculated yeast populations were measured by serial dilutions in YPD (yeast extract peptone, dextrose agar; 10:20:20 g/L), YGC (yeast extract, glucose, chloramphenicol agar; 5:20:0.2 g/L), lysine medium (Oxoid, Hampshire, UK), and CHROMagar™ Candida (Conda, Barcelona, Spain). Samples were taken from the fermentations on days 0, 1, 4, 6 and 8, and on day 8 after sampling, *Saccharomyces* was inoculated. A lysine medium was selected for non-*Saccharomyces* and species identification was done in the differential chromogenic medium. With this method, a rigorous count was achieved in the inoculations.

### 2.6. Oenological Parameters 

The equipment OenoFoss (FOSS Iberia, Barcelona, Spain) using Fourier-transform infrared spectroscopy (FTIR) was used to identify and quantify major compounds such as residual sugars, organic acids, total acidity (as tartaric acid) and volatile acidity (as acetic acid). This technique also estimates pH value. Lactic acid was measured enzymatically using a Y25 enzymatic autoanalyser (Biosystems, Barcelona, Spain). The absorbance at 280, 420, 520 and 620 nm was determined using an Agilent 8453 spectrophotometer (Agilent Technologies S.L., Madrid, Spain) and a 1 mm path length glass cuvette. The pH of each sample was measured with a Crison micropH 2000 pH meter (HACH LANGE, Barcelona, Spain).

### 2.7. Analysis of Fermentative Volatile Compounds Using GC-FID

An Agilent Technologies 6850 gas chromatograph (Network GC System) coupled to a flame ionisation detector (GC-FID) was used for this analysis, as described in [18,19]. Samples were injected after filtration through 0.45 μm cellulose methyl ester membrane filters (Phenomenex, Madrid, Spain). The column used was a DB-624 column. The total run time for each sample was 40 min. The carrier gas used was hydrogen and the internal standard (4-methyl-2-pentanol, 500 mg/L) (Fluka Chemie GmbH, Buchs, Switzerland). The detection limit was 0.1 mg/L. The volatile compounds analysed with this technique were pre-calibrated with five-point calibration curves (r^2^) and all compounds had an r^2^ > 0.999, except 2, 3-butanediol (0.991) and phenylethyl alcohol (0.994).

### 2.8. Determination of Anthocyanins

The following anthocyanins and pyranoanthocyanins were determined using an Agilent Technologies (Santa Clara, CA, USA) series 1200 HPLC chromatograph [20]. Concentrations were calculated with a calibration curve of malvidin-3-O-glucoside (r^2^ = 0.9999, LOD 0.1 mg/L). Injection volume 50 µL. Gradients of solvent A (water/formic acid, 95:5 *v/v*) and B (methanol/ formic acid, 95:5 *v/v*) were used in a reverse-phase Poroshell 120 C18 column (Agilent Technologies, Santa Clara, CA, USA). The detection limit was 0.1 mg/L.

### 2.9. Sensory Analysis

A panel of nine experienced tasters (aged between 25 and 60 years; four women and five men) evaluated the wines that had been bottled and refrigerated for three months. The blind tasting took place in the tasting room of the Department of Chemistry and Food Technology of the Universidad Politécnica de Madrid. The room was equipped with fluorescent lighting and the samples were presented in a random order. The wines (30 mL/tasting glass) were served at 12 ± 2 °C in standard odourless tasting glasses. A glass of water was also provided to the panellists to clean the palate between samples. Before the generation of a consistent terminology by consensus, three visual attributes, seven for aroma and four for taste, were chosen to describe the wines. Panellists used a scale of 1 to 5 to rate the intensity of each attribute. Low values comprised “non-perceptible attributes” and, in contrast, high values featured “strongly perceptible attributes”. Each panellist also evaluated the overall impression, taking into account olfactory and gustatory aspects, as well as the lack of defects. The tasting sheets also had a final blank space for any additional comments or observations on sensory notes or nuances not previously included as attributes.

### 2.10. Statistical Analysis

Statgraphics v.5 software (Graphics Software Systems, Rockville, MD, USA) was used to calculate means, standard deviation, and analysis of variance (ANOVA) and least significant differences (LSD). Significance was set at *p* < 0.05 for the ANOVA matrix F value on the results of the sensory analysis. All treatments were evaluated in triplicate.

## 3. Results

### 3.1. Effects of PEF on the Extraction of Grenache Must after 24 h of Maceration

Table 1 shows the oenological parameters of the must obtained from grapes treated by PEF. These three values, the pH, the °Brix and the total acidity of the must are typical values for Grenache grapes and are not affected by the PEF treatment [3,21]. However, the electroporation of the grape must promoted the extraction of polyphenols from grape skins during maceration. The colour intensity, anthocyanin content, total polyphenolic index and tannin content of the must macerated with the crushed grape treated by PEF was 1.9, 2.2, 1.6 and 2 times higher than those in the must macerated with untreated crushed grape. Values of these indexes for the must macerated with PEF-treated grape must was in the range of some young wines, despite both the maceration temperature and time being much lower and shorter than those used for obtaining young wines [3,5,21,22].

### 3.2. Antimicrobial Effect of PEF and Evolution of the Inoculated Population

The musts were analysed before inoculation with selected non-*Saccharomyces* and *Saccharomyces* yeasts as a control to see the effects of PEFs. It was observed that the microbial load of the PEF-treated must with *Saccharomyces* and non-*Saccharomyces* PEF, which had received 17.5 kV/cm, was 3-log, while the untreated must had a microbial load of 5-log *Saccharomyces* and 4-log non-*Saccharomyces* (Figure 3), verified with different culture media.

Photos and monitoring of the population were taken once the musts were inoculated with the different yeasts (Figure 3) to study their implantation. The photos show the evolution of the population on days 1 and 8 after inoculation with the selected Sc, always using the same logarithm range, and also the differences in yeast types by colour and size, showing differences between inocula, untreated and PEF-treated must. For the untreated must (Figure 3A), we inoculated a *Saccharomyces* yeast and on day 1 we could already see differences between one must and another. In the PEF-treated must the *Saccharomyces* population was at 7-log and the non-*Saccharomyces* had been reduced to 2-log, while in the untreated must the *Saccharomyces* were 7-log and the non-*Saccharomyces* were 5-log. From that day on, the population of non-*Saccharomyces* began to decrease drastically due to the exponential fermentation of *Saccharomyces*, which reached a population of 8-log on day 4. The must inoculated with Lt (Figure 3B) has smaller colonies than the *Saccharomyces* colonies and its colour evolves from cream to pink. Rapid implantation of Lt is observed in both types of must on day 1, reaching 7-log; in the must with PEF the implantation of *Saccharomyces* was weaker. In the must inoculated with Hv (Figure 3C), which in the photos has an intense fuchsia peak in each colony, similar implantation to Lt with 7-log was observed on day 1, although from day 5 there was a progressive decrease due to the increase in the population of *Saccharomyces* and the depletion of nutrients. As for the must with Td yeast (Figure 3D), the colour of the colonies is creamy-yellowish and a good implantation is seen but with a slight difference in that the evolution in the control has a greater population of wild *Saccharomyces* than Td on day 8. The colonies of the must inoculated with Mp (Figure 3E) are slightly larger than the rest and have a creamy-whitish colour. A good implantation was seen but, in both cases, once the wild *Saccharomyces* increased their population to a great extent, they displaced the Mp, being clearly seen from day 4 of fermentation. The must inoculated with Sp yeast (Figure 3F), whose colonies are smaller and pale pink in colour, showed a worse implantation (6-log) and a better evolution and growth in the PEF-treated must.

### 3.3. Oenological Parameters

In the fermentation kinetics (Figure 4) of the must, differences in fermentative power between yeasts were observed. On day 4 of fermentation, there were three differentiated groups of yeasts: on the one hand, the musts inoculated with Sc had the highest alcoholic strength, which was between 12.4 ± 0.00 and 12.8 ± 0.21% *v/v* being superior to the one that had received PEF treatment; in the second group, there were the inoculums with PEF-treated Hv, Td and untreated inoculums Lt, Hv, Td, Mp and Sp, with an alcoholic degree between 8.5 ± 0.25 and 10.5 ± 0.20% *v/v*; finally, the third group comprised the inoculums with PEF-treated musts of Lt, Mp and Sp, with an alcoholic degree between 5.2 ± 0.64 and 7.4 ± 0.35% *v/v*. On day 6, the musts inoculated with Sc had almost finished fermentation; a second group were between 12.2 ± 0.15 and 13.5 ± 0.20% *v/v*, and those with a slower fermentation were between 9.9 ± 0.06 and 10.5 ± 0.21% *v/v*. On day 8, just before inoculating the Sc to facilitate the completion of fermentation, only three musts that had received PEF treatment were below 14 degrees alcohol (Lt, Mp and Hv). Once the Sc was inoculated, the musts increased their alcohol content, although the must with Hv PEF could not finish fermentation and the fermentation with Lt PEF also had a few grams of residual sugars.

For a better understanding of the oenological data in Table 2, the fermentations with the Sc inoculum had very similar parameters throughout the fermentation. The PEF-treated must with the Lt inoculum was very different from the untreated must: the PEF-treated must with the Lt had a lower alcohol content (14.7 ± 0.1 % *v/v*), a higher total acidity (5.37 ± 0.29 g/L), and a significantly lower pH (3.19 ± 0.09). The same path followed the fermentation with PEF-treated must with the Hv, where it was seen that it could not finish fermentation (14.37 ± 0.12% *v/v*) and ended with a residual sugar of 15.73 ± 0.7 g/L. With respect to Td, Mp and Sp, all were slower on day 8 of fermentation than those that had received PEF, but all finished with fairly similar parameters. It should be noted that all untreated must had lower sugars on day 8 of fermentation due to the rapid implantation of wild yeasts.

### 3.4. Lactic and Malic Acid

At the end of fermentation, the high production of lactic acid (Figure 5A) was observed in PEF-treated must with Lt, giving 3.62 ± 0.84 g/L, while in the untreated must with Lt it only reached 0.80 ± 0.09 g/L. On the other hand, the malic acid (Figure 5B) obtained can be grouped in three groups from day 4, becoming more noticeable from day 8: in the first group, in the upper part, six musts are seen with a malic acidity on day 20 between 0.98 ± 0.04 and 1.29 ± 0.09 g/L; in the second group, in the middle of the figure, the data are between 0.55 ± 0.04 and 0.83 ± 0.05 g/L; finally, the third group in the lower part are the Sp yeasts giving 0.10 ± 0.02 and 0.00 ± 0.00 g/L, which are always below the one that had received PEF.

### 3.5. Fermentative Volatiles

To evaluate the impact of these non-*Saccharomyces* yeasts on the PEF-treated and untreated must, we focused on three different categories: higher alcohols, carbonyl compounds, and total esters. Analyses were carried out on days 8 and 20 to evaluate their evolution and to compare between the two musts/wines. In the higher alcohols on day 8 of fermentation before inoculation of Sc (Figure 6A), it was observed that except for Sc PEF (348.07 ± 20.37 mg/L) and Sp PEF (243.65 ± 13.89 mg/L), all the other fermentations exceeded 350 mg/L, which can generate undesirable wine aromas, and in all cases, the higher alcohols in the PEF-treated musts were lower or equal. These high levels were mainly produced by 3-methyl-1-butanol, which in no case exceeded 300 mg/L, and by isobutanol, which was in all musts between 75 and 125 mg/L. It is interesting to note that 1-propanol in the must inoculated with Lt untreated and PEF-treated almost doubled its amount compared to the others, having between 79.84 ± 3.39 and 74.85 ± 5.20 mg/L.

The carbonyl compounds (Figure 6A), which include acetoin and diacetyl, were slightly higher in those that were PEF-treated, especially with yeast Hv that generated 30.63 ± 3.73 mg/L of acetoin. 

The total esters (Figure 6A), which include ethyl acetate, isobutyl acetate, ethyl butyrate, ethyl lactate, isoamyl acetate and 2-phenylethyl acetate, showed that those that were PEF-treated had less ethyl acetate than the untreated ones, the must with the highest amount (59.80 ± 4.13 mg/L) being the Sp untreated one. In all the other fermentations the ethyl acetate was between 15 and 35 mg/L. The concentration of 2-phenylethyl acetate was much higher in the Lt untreated and PEF-treated must, being 28.71 ± 0.77 and 88.42 ± 10.14 mg/L respectively.

On the other hand, on day 20, at the end of fermentation, it was observed that the higher alcohols (Figure 6B) remained elevated except for Sp PEF with 218.28 ± 14.26 mg/L, which did not increase. Carbonyl compounds decreased in general although the PEF-treated must with Hv (11.63 ± 1.68 mg/L) was still higher than the rest. Finally, the esters decreased in general, all being below 72 mg/L, although it should be noted that 2-phenylethyl acetate was twice as high (14.64 ± 1.31 mg/L) in the PEF-treated must with Hv compared to the rest of the yeasts and treatments, and ethyl lactate in the Lt PEF was 47.19 ± 4.00 mg/L, the highest in all cases. It is worth mentioning that 3-ethoxy-1-propanol was higher in the Td PEF on day 8 (8.05 ± 1.61 mg/L) than in the control (2.92 ± 0.72 mg/L). In both cases, at the end of fermentation, this compound decreased; in the control it disappeared completely and in the PEF it decreased to 3.17 ± 0.58 mg/L.

### 3.6. Anthocyanins and Colour

The typical colour of red wines is due to the presence of anthocyanins and their derivatives. Figure 7A shows the total anthocyanin content of grapes. This fraction includes acylated and non-acylated anthocyanins, and anthocyanins acylated from free anthocyanins with acetic and p-coumaric acid. In general, total anthocyanins were always higher in wines with PEF-treated must, except for the coumaroylate fraction. The mean of the glucosylated fraction in the PEF-treated must was 202.17 ± 2.49 mg/L while in the untreated must it was 196.97 ± 0.83 mg/L, with malvidin-3-glucoside being the main anthocyanin, representing practically the totality of monomeric anthocyanins in all wines. The mean of the acetylated fraction was 4.55 ± 0.11 mg/L in the PEF-treated must and 4.35 ± 0.08 mg/L in the untreated must, and the p-coumaroylated fraction was 12.91 ± 0.70 mg/L in the PEF-treated must and 13.67 ± 0.19 mg/L in the untreated must. Concerning the vitisins obtained (Figure 7B), on the one hand the mean of vitisins B was higher in those that were PEF-treated (0.47 ± 0.03 mg/L), while the mean of vitisins A was also higher in those that were PEF-treated (2.46 ± 0.09 mg/L), the highest value being that of the yeast Sp PEF with 4.3 ± 0.15 mg/L. It was observed (Figure 7C) that the wines that were PEF-treated and untreated with Sc produced vinyl epicatechins (0.18 ± 0.01 and 0.13 ± 0.00 mg/L) lower amounts were observed in the untreated Lt and Sp yeasts and untreated and PEF-treated Mp.

Colour intensity, hue and total polyphenols index (TPI) were determined at days 0 and 20 (Table 3) to evaluate colour changes. A slightly higher IPT was observed in the musts with PEF, as well as slightly lower colour intensity. On the other hand, the highest colour intensity in the wines was in both cases for the Lt yeast, due to the effect of a lower pH. The latter was higher in the control, and the Hv and Mp control were the highest in terms of hue. Finally, it was observed that the mean IPTs in both cases were practically the same, PEF 27.14 ± 0.28 and control 27.24 ± 0.36.

### 3.7. Sensory Analysis

The wines were sensorially evaluated for possible differences (Figure 8). In general there were no major differences between the PEF treated and untreated, however, tasters detected a greater colour intensity in the Sc PEF (4.00 ± 0.71) along with a greater hue (2.78 ± 0.83), perhaps due to greater oxidation; on the other hand, the lowest hue, i.e., bluer tones, was found in the Sp PEF with 1.67 ± 0.87. Turbidity was minimal since an effective and efficient filtering process was carried out. In general, in terms of aromatic intensity and quality, the wines with PEF-treated scored higher. The wines were slightly herbaceous, a bit floral and with a medium level of fruitiness, the lowest being the Mp untreated wines (1.67 ± 0.87) and the highest the Sp PEF (3.11 ± 1.36). It was seen that the most oxidised samples were those fermented with Sc, perhaps due to faster fermentation. Although the Hv PEF was the one with the highest residual sugar, it was not significantly detected by the tasters; however, the acidity parameter was detected in both cases with the Lt yeast and Lt untreated yeast with 3.56 ± 0.88 and Lt PEF 4.00 ± 0.50. In the overall perception of taste, Sp PEF (3.22 ± 1.09) was superior to the rest, although with a greater deviation. However, as mentioned above, there were no major differences, but overall the tasters found the wines with PEF to be more complex, with more aromatic intensity, more fruit and acidity, all due to better implantation of the non-*Saccharomyces* yeasts that generate a slight increase of desirable metabolites in the wines.

For a complete understanding of the sensory analysis (Appendix A), an ANOVA statistical analysis was performed separating by each variability factor: must treatment with PEF/no must treatment and fermentative biotechnology (6 yeasts evaluated).

It was observed that in colour intensity there were no significant differences due to pretreatment of the must with PEF, but there were significant differences depending on the fermentative biotechnology used. In terms of hue, there were significant differences due to pretreatment of the must with PEF in the Sp yeasts. It was also observed that in the fruit parameter there were significant differences due to pretreatment of the must with PEF in the Sp yeast and also significant differences depending on the fermentation biotechnology used. Oxidation had no significant differences due to pretreatment of the must with PEF, but did have significant differences depending on the fermentation biotechnology used. In the sweetness parameter, there were no significant differences due to pretreatment of the must with PEF, but there were significant differences depending on the fermentation biotechnology used. Finally, there were no significant differences in acidity due to pretreatment of the must with PEF, but there were significant differences depending on the fermentation biotechnology used.

## 4. Discussion

In this investigation, the ability of PEF to improve mass transfer and for use in microbial decontamination at lower temperatures than those used in thermal processing was used to investigate the effect of non-*Saccharomyces* starters in the oenological and sensorial characteristics of red wine. In the first step, PEF treatment was applied to electroporate the cells of the grape skins to accelerate the extraction of polyphenols. This effect has been widely described in the literature. However, most of the studies that have demonstrated the effect of PEF in improving polyphenol release have been conducted applying treatments of low energy (<10 kJ/kg) and extending maceration for several days at room temperature [23]. In this investigation, according to results previously obtained by [21], the total specific energy of the treatment was significantly increased to obtain a large degree of electroporation that resulted in a high concentration of polyphenols in the must within a few hours of maceration at 5 °C (this temperature is optimal to keep the must and the proliferation of micro-organisms to a minimum).

Red must obtained from crushed grape treated by PEF was decontaminated by applying a second PEF treatment at a higher electric field strength (17.5 kV/cm) than those applied to the grape must (5 kV/cm). It is well known that, as microbial cells are smaller than cells of plant tissues, the external electric field required to achieve the critical transmembrane potential for manifestation of electroporation is higher [24]. Several studies have reported that PEF could be an alternative to the addition of SO_2_ due to its effectiveness to inactivate bacteria and yeast in the must of white grapes in winemaking [25].

It was observed in the population counts made on plates that in all cases the implantation of selected non-*Saccharomyces* yeasts was better in the must that had received PEF, due to the inactivation of part of their wild yeasts [12,26]. 

Due to this differentiation in yeast implantation, the fermentations that had received PEF were slightly slower, particularly Hv, probably due to the lack of nutrients such as thiamine [27,28] and also Lt due to a considerable increase in acidity [14,29]. Volatile acidity with Td yeast was the lowest in both cases, as also seen in other studies [30,31,32]. It should be noted that Lt yeast is capable of producing large amounts of lactic acid depending on different factors [29,33]. This amount of lactic acid was produced around day 6, as shown in previous studies [14,34] and was much higher in the must that had received PEF due to its better implantation. On the other hand, the Sp yeast in both treatments reduced the amount of malic acid to zero (with a greater reduction in the Sp PEF due to its better implantation) [35] and, in general, it was observed that the non-*Saccharomyces* yeasts that had received PEF had produced less malic acid.

As for volatile compounds, each yeast contributes its metabolites, which are significant in improving wine complexity [36,37,38]. On the one hand, the higher alcohols on both days 8 and 20 of fermentation were in all cases, except Sp PEF, above the perception threshold, which can cause fusel aromas and even irritants [39]. The predominant alcohol, 2-methyl 1-butanol, in no case exceeded the perception threshold; however, isobutanol in Td and Mp yeasts with PEF on day 20 was within the perceptible taste threshold of 75 mg/L [40]. As in previous articles, the 1-propanol production was always higher at the end of fermentations in the inoculations with Lt PEF compared to the other yeasts and the other untreated wine on days 8 and 20, being more accentuated the higher the amount of lactic acid produced. A possible cause could be the small-scale fermentations; it was reported that it seems that at a large scale, this compound decreases [14,41]. To continue, carbonyl compounds, with buttery and caramel aromas, have a low perception threshold; in the case of diacetyl, no fermentation reached this threshold, and though it could be detected in acetoin, this was very dependent on the grape variety [42,43]. Finally, esters are closely related to readily assimilable nitrogen (YAN) [44] and to fermentation time. In no case did ethyl acetate exceed 60 mg/L to produce undesirable varnish aromas [41]. Ethyl lactate was much higher on day 20 than the rest in the Lt PEF. This is a direct consequence of the strong lactic acid production by most Lt strains and coincides with recent research developed on the influence of Lt on the fermentative aroma of wine [34,45], although it remains far from the perception threshold [46]. Both isobutyl acetate and isoamyl acetate were slightly above the perception threshold, with tropical fruit aromas in all cases [47], and 2-phenylethyl acetate was higher in Hv PEF. This is a specific characteristic of this yeast and it has been seen that with better implantation of the yeast, this aromatic ester with rose odour is higher [48,49]. As in previous studies, Td PEF produced 3-ethoxy-1-propanol, which was lost during fermentation but somewhat conserved on day 20 [17].

Research shows that the application of low-energy PEF (<10 kJ/Kg) can accelerate the extraction of polyphenols [3,50]; however, as there were no skins in the second PEF treatment, the small difference in anthocyanins seen between the PEF-treated wine and the untreated wine are due to the characteristics of each yeast. The most significant result was in vitisin A with Sp PEF, which is a specific property of this yeast, and at the same time, this yeast was one of those with the lowest amount of total anthocyanins [51]. It was also seen that Sc in both cases produced more vitisin B than the rest due to a higher amount of acetaldehyde at the end of fermentation (a precursor of vitisin B) [52]. Vinylphenolic pyranoanthocyanins (VPA) are formed due to the enzymatic activity hydroxycinnamate decarboxylase (HCDC) of the Sc species and it has also been seen that certain strains of Sp can produce it [53], but in our case, both Lt control, Sp control and both Mp were due to better and faster implantation of wild yeasts. Regarding wine colour, the lower colour intensity in the grapes fermented with Sc PEF is potentially associated with the formation of insoluble polymeric pigments [54]. The second round of PEF was performed on must without skins, resulting in a lower extraction of colour and pigments.

Finally, in the overall wine sensory analysis, the different yeasts and treatments were able to generate very different and complex wines with almost no defects compared to the pure culture of *S. cerevisiae*. Tasters were generally better able to notice the characteristics of each yeast in the musts treated with PEF due to better implantation of the selected microbiota. Greater colour intensity was noted in the wines without PEF, which is consistent with the data provided by the spectrophotometer. The hue in the wine with Lt PEF was one of the lowest due to the protection exerted by the high acidity produced by the yeast in that wine [14]. As for the fruit that the tasters found in the wines, they agreed that they found more in the wines with Lt and Sp PEF since they were among those with the most esters. In general, no differences were detected between the treated and untreated wines [55], except for the Sp yeast; the changes were due to the better or worse implantation of the selected yeast, although they were not very noticeable either, which means that the PEF treatment did not sufficiently decrease the indigenous population for optimal or full expression of the metabolites specific to each yeast.

## 5. Conclusions

The use of PEF improves polyphenol extraction with a few hours of maceration and decreases wild microbiota to improve the implantation of selected non-*Saccharomyces* yeasts, which improves the qualities of each yeast by having a lower competitiveness between yeasts, such as wine acidity, production of fermentative volatiles including esters, and formation of derived anthocyanins, particularly vitisin A. The combined use of emerging nonthermal technologies, such as PEF and non-*Saccharomyces* yeast fermentation, improves the organoleptic profile of wines and reduces the use of sulphites.

## Figures and Tables

**Figure 1 foods-10-01472-f001:**
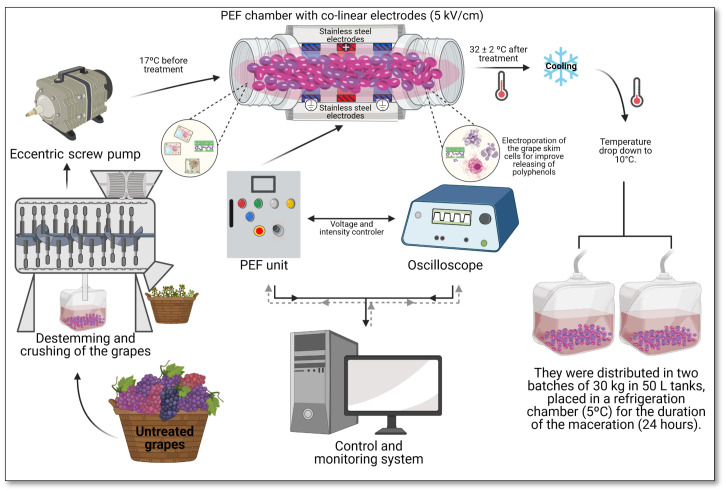
Schematic of pulsed electric fields PEF processing of grapes.

**Figure 2 foods-10-01472-f002:**
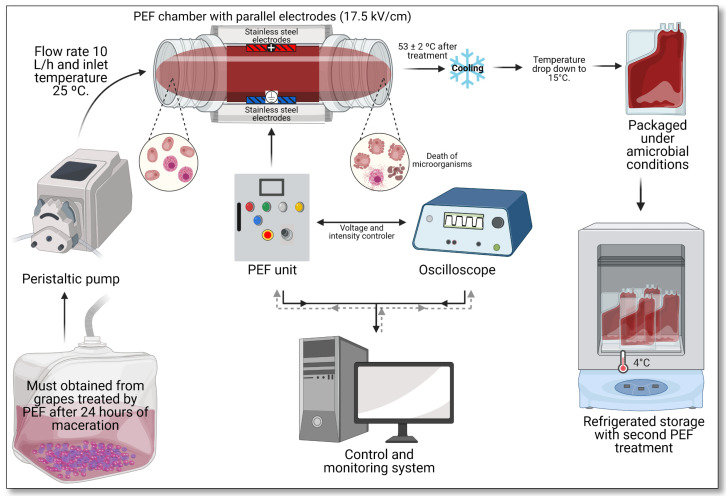
Schematic of the PEF processing of must.

**Figure 3 foods-10-01472-f003:**
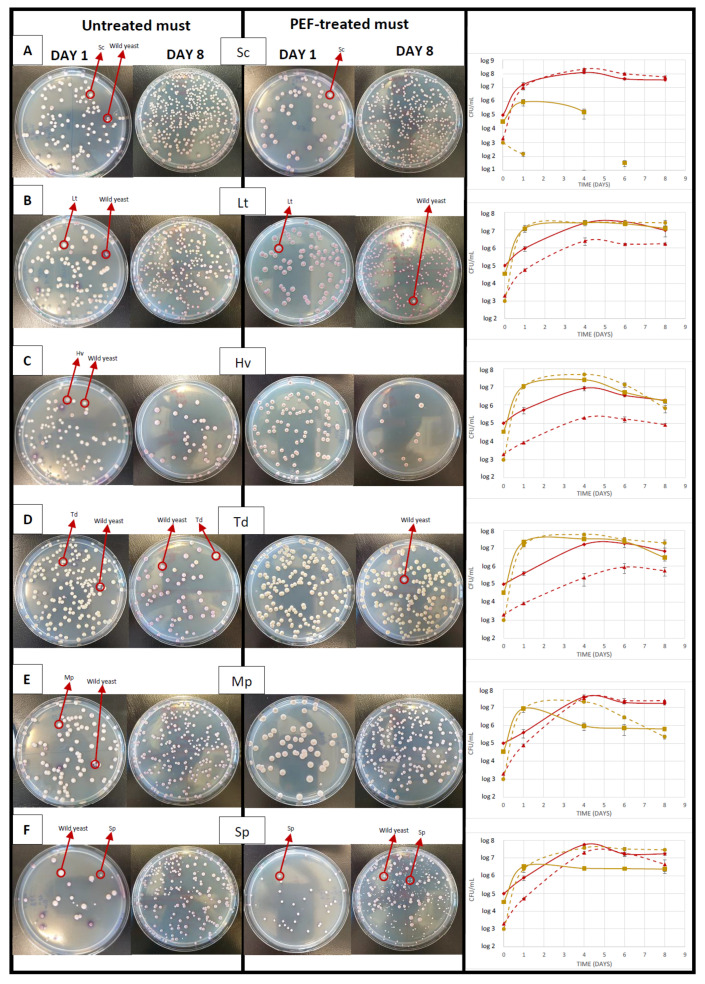
Days 1 and 8 plates of the different fermentations with and without PEF and CFU/mL counts throughout fermentation of untreated must with *Saccharomyces* (solid red line) and non-*Saccharomyces* yeasts (solid yellow line), in PEF-treated must with *Saccharomyces* yeasts (dashed red line) and non-*Saccharomyces* yeasts (dashed yellow line). Musts were inoculated with (**A**) *S. cerevisiae*, (**B**) *L. thermotolerans*, (**C**) *H. vineae*, (**D**) *T. delbrueckii*, (**E**) *M. pulcherrima*, and (**F**) *S. pombe*. Values are means with standard deviations (error bars) from three independent triplicates.

**Figure 4 foods-10-01472-f004:**
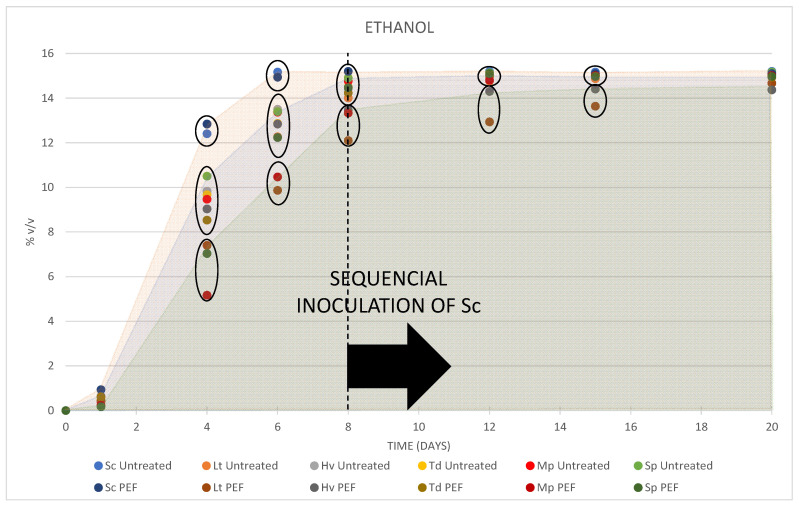
Evolution of the alcoholic strength produced during the fermentation of the different inoculations. The three colours in the figure show the fermentative power before sequential inoculation of the Sc.

**Figure 5 foods-10-01472-f005:**
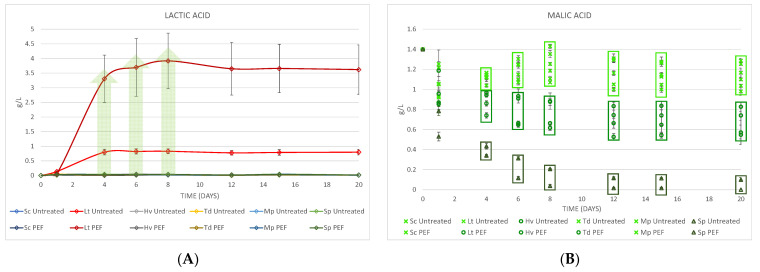
(**A**) Lactic acid produced in the fermentations. (**B**) Malic acid at the end of fermentation divided into three groups by amount of malic acid produced. Values are means ± sd (*n* = 3).

**Figure 6 foods-10-01472-f006:**
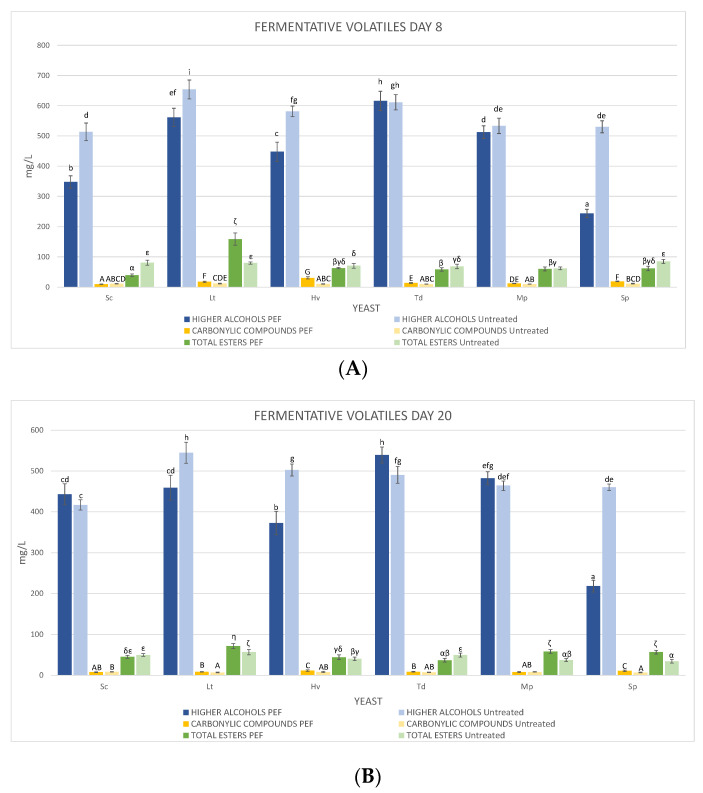
Fermentative volatiles in wines obtained from untreated and PEF-treated musts/wines. (**A**) Day 8 and (**B**) day 20. Values are means ± sd (*n* = 3). A different letter (lowercase, capital and Greek letters) for the same category of volatiles means significant differences (*p* < 0.05).

**Figure 7 foods-10-01472-f007:**
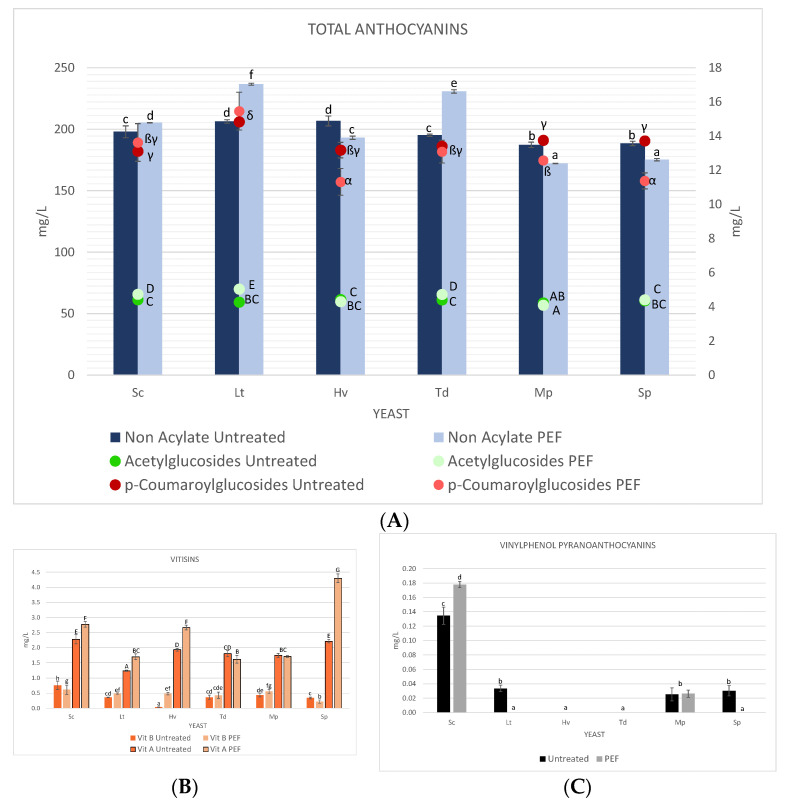
(**A**) Total anthocyanins divided into three groups of untreated and PEF-treated wine. (**B**) Vitisins A and B. (**C**) Vinylphenol pyranoanthocyanins. Values are means ± sd (*n* = 3). A different letter (lowercase, capital and Greek letters) for the same category of anthocyanins means significant differences (*p* < 0.05).

**Figure 8 foods-10-01472-f008:**
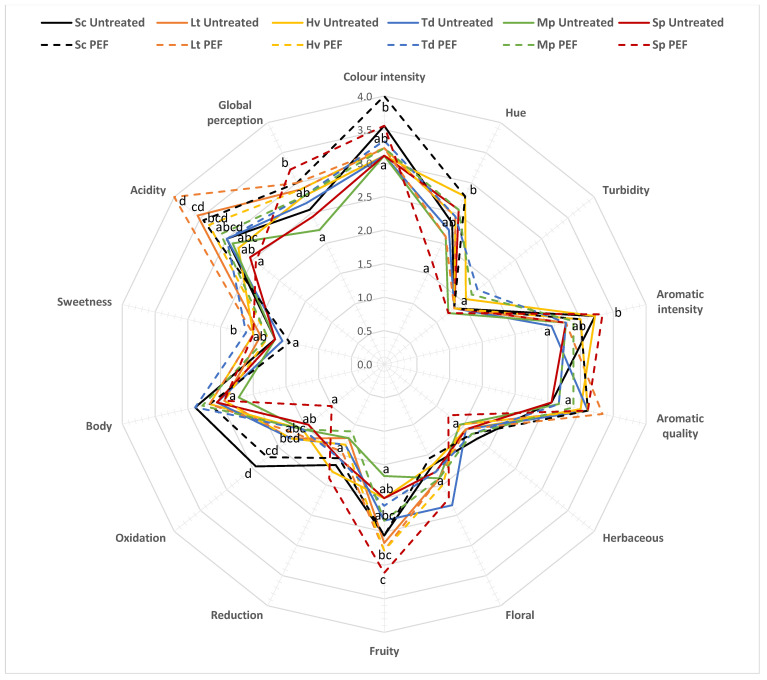
Sensory analysis of wines. The values are the average from nine tasters. The same attributes with the same letter are not significantly different (*p* < 0.05).

**Table 1 foods-10-01472-t001:** Oenological parameters of the Grenache must after 24 h of maceration at 5 °C with grapes treated by PEF.

Oenological Parameters	
pH	3.65 ± 0.02
Brix (g/L)	25.5 ± 0.21
Total acidity (g/L)	4.27 ± 0.08
Colour intensity	17.64 ± 0.10
Total phenol index (280 nm)	45.3 ± 0.28
Anthocyanins (mg/L)	522.3 ± 24.0
Tannins (mg/L)	104.5 ± 13.8

**Table 2 foods-10-01472-t002:** Oenological parameters of the different fermentations before sequential inoculation with Sc (day 8) and on the last day of analysis (day 30). Values are means with standard deviations, n = 3. Values with the same letter in the same column are not significantly different (*p* < 0.05).

Parameters	Ethanol (% *v/v*)		Total Acidity (g/L)		Volatile Acidity (g/L)		Glucose/Fructose (g/L)		pH	
**Yeast/Days**	8	20	8	20	8	20	8	20	8	20
Sc Untreated	15.20 ± 0.00 g	15.20 ± 0.10 f	4.33 ± 0.06 e	3.87 ± 0.06 c	0.07 ± 0.00 a	0.09 ± 0.01 a	0.30 ± 0.00 a	0.13 ± 0.23 ab	3.38 ± 0.01 cd	3.55 ± 0.01 def
Sc PEF	15.20 ± 0.10 g	15.10 ± 0.10 d ef	4.37 ± 0.06ef	3.93 ± 0.06 cd	0.08 ± 0.02 ab	0.09 ± 0.01 ab	0.00 ± 0.00 a	0.00 ± 0.00 a	3.40 ± 0.01 de	3.55 ± 0.01 def
Lt Untreated	14.00 ± 0.00 c	14.97 ± 0.06 cd	4.90 ± 0.10 g	4.77 ± 0.06 f	0.17 ± 0.01 e	0.15 ± 0.01 c	23.60 ± 1.90 e	0.30 ± 0.36 ab	3.30 ± 0.02b	3.38 ± 0.01 b
Lt PEF	12.10 ± 0.10 a	14.70 ± 0.10 b	5.50 ± 0.17 h	5.37 ± 0.29 g	0.33 ± 0.03 h	0.34 ± 0.06 e	51.10 ± 0.87 g	3.47 ± 0.85 f	3.13 ± 0.02 a	3.19 ± 0.09 a
Hv Untreated	14.43 ± 0.15 e	15.03 ± 0.12 cde	4.00 ± 0.00 bc	3.93 ± 0.06 cd	0.25 ± 0.02 g	0.23 ± 0.02 d	18.17 ± 3.25 d	0.80 ± 0.17 bc	3.44 ± 0.01 fg	3.49 ± 0.00 c
Hv PEF	13.47 ± 0.15 b	14.37 ± 0.12 a	3.63 ± 0.15 a	3.37 ± 0.06 a	0.31 ± 0.02 h	0.31 ± 0.01 e	33.00 ± 2.52 f	15.73 ± 0.70 g	3.49 ± 0.02 h	3.58 ± 0.01 ef
Td Untreated	14.50 ± 0.10 e	15.03 ± 0.06 cde	4.33 ± 0.06 e	4.13 ± 0.06 e	0.11 ± 0.02 cd	0.11 ± 0.00 ab	13.33 ± 0.45 c	1.90 ± 0.35 de	3.42 ± 0.02 ef	3.47 ± 0.01 c
Td PEF	14.23 ± 0.15 d	14.93 ± 0.06 c	3.90 ± 0.17 b	3.60 ± 0.17 b	0.12 ± 0.02 cd	0.11 ± 0.01 ab	16.80 ± 3.02 cd	2.13 ± 0.21 e	3.47 ± 0.02 h	3.59 ± 0.02 f
Mp Untreated	14.77 ± 0.06 f	15.10 ± 0.00 def	4.50 ± 0.00 f	4.20 ± 0.00 e	0.13 ± 0.01 d	0.12 ± 0.01 abc	8.90 ± 1.55 b	1.23 ± 0.87 cd	3.36 ± 0.01 c	3.45 ± 0.01 c
Mp PEF	13.33 ± 0.06 b	15.00 ± 0.10 cd	4.13 ± 0.06 cd	4.10 ± 0.00 de	0.21 ± 0.01 f	0.15 ± 0.01 c	32.07 ± 0.93 f	0.80 ± 0.44 bc	3.40 ± 0.01 de	3.49 ± 0.00 c
Sp Untreated	14.90 ± 0.10 f	15.17 ± 0.12 ef	3.70 ± 0.00 a	3.37 ± 0.06 a	0.10 ± 0.01 bc	0.12 ± 0.01 bc	7.07 ± 4.80 b	0.00 ± 0.00 a	3.45 ± 0.01 g	3.54 ± 0.01 d
Sp PEF	14.47 ± 0.06 e	14.97 ± 0.12 cd	4.17 ± 0.06 d	3.63 ± 0.06 b	0.22 ± 0.02 f	0.21 ± 0.02 d	9.40 ± 1.78 b	0.03 ± 0.06 ab	3.40 ± 0.02 de	3.54 ± 0.00 de

**Table 3 foods-10-01472-t003:** Parameters of colour intensity, tonality and total polyphenols index (TPI) measured before inoculation (day 0) and on the last day of fermentation (day 20). Values are means ± sd (*n* = 3). A different letter for the same day means significant differences (*p* < 0.05).

Parameters	Colour Intensity		Tonality		TPI	
Yeast/Day	Day 0	Day 20	Day 0	Day 20	Day 0	Day 20
Sc Untreated	17.64 ± 0.10 b	6.02 ± 0.26 d	6.33 ± 0.10 a	5.52 ± 0.30 c	45.9 ± 0.10 a	27.3 ± 0.82 cdef
Lt Untreated		6.94 ± 0.20 e		4.93 ± 0.27 a		24.7 ± 0.49 a
Hv Untreated		5.99 ± 0.15 d		6.44 ± 0.18 h		27.7 ± 0.31 def
Td Untreated		5.74 ± 0.12 bcd		5.70 ± 0.14 de		27.2 ± 0.35 cde
Mp Untreated		6.81 ± 0.21 e		6.59 ± 0.26 h		28.8 ± 0.12 g
Sp Untreated		5.71 ± 0.17 bcd		5.46 ± 0.25 bc		27.0 ± 0.10 c
Sc PEF	16.92 ± 0.20 a	5.05 ± 0.05 a	6.32 ± 0.20 a	5.31 ± 0.06 b	47.9 ± 0.20 b	26.1 ± 0.31 b
Lt PEF		6.09 ± 0.08 d		5.54 ± 0.08 cd		27.1 ± 0.39 cd
Hv PEF		5.88 ± 0.07 cd		6.15 ± 0.08 g		28.4 ± 0.39 fg
Td PEF		5.41 ± 0.03 ab		5.57 ± 0.04 cde		26.1 ± 0.29 b
Mp PEF		5.75 ± 0.25 bcd		5.72 ± 0.29 e		27.8 ± 0.11 ef
Sp PEF		5.46 ± 0.22 abc		5.97 ± 0.26 f		28.0 ± 0.22 f

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
