# Peer review of "Pulsed Electric Fields to Improve the Use of Non-Saccharomyces Starters in Red Wines"

_foods, 2021, doi:10.3390/foods10071472_

Round 1

Reviewer 1 Report

In what way and to what extent the efficacy of PEF treatments are affected by the ionic strength, electrical conductivity, ionic composition of the treated medium?

In light of compositional heterogeneity of the must (batch to batch, seasonality, etc) what are the anticipated implications of the application of PEF in winemaking?

Author Response

Thank you very much reviewer 1 for taking the time to read and understand our article and we will be pleased to answer your questions.

- In what way and to what extent the efficacy of PEF treatments are affected by the ionic strength, electrical conductivity, ionic composition of the treated medium?

- The efficacy of PEF mainly depends on the electric field strength applied, treatment time (number of pulses x pulse width) and total specific energy applied. Conductivity and the related parameters such as ionic strength and ionic composition affect to the requirements of the PEF generator to apply a certain PEF treatment at a given intensity rather than to the efficacy of the treatment. Conductivity affects to the resistance of the treatment chamber and consequently to the current intensity that it is required to apply a PEF treatment to achieve a given electric field strength. The higher the conductivity of the treatment medium, the lower the resistance of the treatment chamber and consequently the more intensity is required to reach a certain electric field. A higher intensity requirement implies that to achieve certain electric field intensity a higher total specific energy is required. Summarizing if PEF treatments of the same intensity are applied to media of different conductivity and ionic composition the same efficacy is obtained.

- In light of compositional heterogeneity of the must (batch to batch, seasonality, etc) what are the anticipated implications of the application of PEF in winemaking?

- AS with other techniques the characteristics of the raw material affect its effectiveness. This is an aspect that has been previously discussed in several articles and reviews.

López, N., Puértolas, E., Condón, S., Álvarez, I., & Raso, J. (2008a). Application of pulsed electric fields for improving the maceration process during vinification of red wine: Influence of grape variety. European Food Research and Technology, 227(4), 1099.

López-Giral, N., González-Arenzana, L., González-Ferrero, C., López, R., Santamaría, P., López-Alfaro, I., & Garde-Cerdán, T. (2015). Pulsed electric field treatment to improve the phenolic compound extraction from Graciano, Tempranillo and Grenache grape varieties during two vintages. Innovative Food Science & Emerging Technologies, 28, 31–39

E Puértolas, N López, S Condón, I Álvarez, J Raso Potential applications of PEF to improve red wine quality Trends in Food Science & Technology 21 (5), 247-255

Pulsed electric fields in wineries: potential applications G Saldaña, E Luengo, E Puértolas, I Álvarez, J Raso Handbook of electroporation. Springer International Publishing AG.

Reviewer 2 Report

Dear authors,

I carefully read your manuscript entitled ‘Pulsed electric fields to improve the use of non-Saccharomyces starters in red wines’.  While the work is of interest, certain improvements are required to improve the readability of the manuscript. Multiple sections require more concise wording and/or explanations as the main points are lost. Moreover, please revisit the statistical approach and the data representation, as it remains unclear to which extent the parameters are affected by i) yeast treatment and ii) PEF treatment for the same/different yeast modality (not only for sensory data as specified below, but throughout the manuscript). Please find more specific comments below.

  1. 61 and thereafter: The term ‘grape mash’ is not commonly used. Please replace or explain.
  1. 77 PEF processing of grapes and L. 99 PEF processing of grapes – It is not clear why all the grapes underwent PEF processing. Are two schemes necessary to show the processes? Integrating into one would make the process clearer. With lots of details, it is difficult to follow the main line
  1. 146 ‘Single scale fermentation’ – Does that mean no replication? How do the authors justify that?
  2. 152 – L. 190 No need for separate paragraphs for each analyte. This should be better organised.
  1. 214 ‘Table 1 shows the oenological parameters of the Grenache must after 24 hours of maceration at 5 °C with grapes treated by PEF. The pH, °Brix, and total acidity of the must correspond to the typical values previously reported for must of Grenache grapes. These results confirm those of other authors that indicate that PEF treatment did not affect these three indexes’ An example of excessively long wording that is present throughout the article. Please shorten the article to improve readability. Also, add reference to support the claim ‘must correspond to the typical values previously reported for must of Grenache grapes’
  1. 219 – L 222 Refer to the data to support this claim.

Table one – What do the values represent?

  1. 229 – L. 233 It is unclear at which stage this analysis occurred. Also, as discussing the results, refer to the table/figure and explain properly.
  1. 234 – 258 and Figure 2; Again, overly wordy paragraph where the main point gets lost. Make it more concise. On the other hand, certain information is missing (remind the reader about the media, the initial inoculation rates etc.). Moreover, the data appears to be showing certain trends but lacks clarity. The yeast identification is still tentative.
  1. 238 Inocula, not inoculums
  2. 244 ‘The must inoculated with Lt (Figure 2b) is smaller’ What does that mean?

Figure 3: Sequential inoculation; not sequencial. What do the ellipses represent?

  1. 389 Sensory results are unclear. Please re-analyse the data so as to clearly show the effect of 1) yeast treatment 2) PEF treatment.

Figure 7: The statistical differences (letters) are not perceivable on the spider plot which is overall too crowded.

  1. 417 It is still unclear why 5 degrees was used for the maceration. Please explain.
  1. 425 – 427. How do you explain the differences in the population of wild yeast counts between the same PEF but different yeast treatment at early stages of fermentation?

Author Response

Thank you very much reviewer 2 for taking the time to read our article carefully, which we appreciate and will try to improve it as you indicate.

  1. 61 and thereafter: The term ‘grape mash’ is not commonly used. Please replace or explain.
  2. Thank you very much for your comment they have all been changed to "crushed grape" which is better understood.
  1. 77 PEF processing of grapes and L. 99 PEF processing of grapes – It is not clear why all the grapes underwent PEF processing. Are two schemes necessary to show the processes? Integrating into one would make the process clearer. With lots of details, it is difficult to follow the main line
  2. Thank you very much for your question, although we understand your point of view, we believe it is necessary to make two explanations, because although the procedure is similar, the first procedure is to obtain more colour in the must with a certain intensity of electric fields and different to the second procedure where the aim is to eliminate the indigenous microbiota and where a higher intensity of electric fields is used.
  1. 146 ‘Single scale fermentation’ – Does that mean no replication? How do the authors justify that?
  2. A single laboratory scale fermentation was carried out with 250 mL flasks but in triplicate (6 different yeasts, 2 treatments and in triplicate, in total 36 flasks) in order to be able to perform statistical analysis correctly.
  1. 152 – L. 190 No need for separate paragraphs for each analyte. This should be better organised.
  2. Thank you very much, we understand that a reduction of the analytes is necessary and we have put together several analyses in one paragraph for a better organisation.
  1. 214 ‘Table 1 shows the oenological parameters of the Grenache must after 24 hours of maceration at 5 °C with grapes treated by PEF. The pH, °Brix, and total acidity of the must correspond to the typical values previously reported for must of Grenache grapes. These results confirm those of other authors that indicate that PEF treatment did not affect these three indexes’ An example of excessively long wording that is present throughout the article. Please shorten the article to improve readability. Also, add reference to support the claim ‘must correspond to the typical values previously reported for must of Grenache grapes’
  2. Thank you very much for your comment, we have shortened the wording and introduced references to justify the results.
  1. 219 – L 222 Refer to the data to support this claim.
  2. Thank you very much for your comment, we have introduced several references to justify the results.
  1. Table one – What do the values represent?
  2. We have therefore added a longer table header for a better understanding of the table.
  1. 229 – L. 233 It is unclear at which stage this analysis occurred. Also, as discussing the results, refer to the table/figure and explain properly.
  2. As for their doubt, the musts were analysed/plated on the same day of the inoculation of the selected yeasts but before their inoculation. It has been added in which figure you can see with which starting population the fermentations started.
  1. 234 – 258 and Figure 2; Again, overly wordy paragraph where the main point gets lost. Make it more concise. On the other hand, certain information is missing (remind the reader about the media, the initial inoculation rates etc.). Moreover, the data appears to be showing certain trends but lacks clarity. The yeast identification is still tentative.
  2. Thank you very much for your point of view, we have reviewed the paragraph and we think it is necessary to be so long, as it explains a figure with many images and some complexity, if we shortened the paragraph the explanations would be too poor to understand the figure. On the other hand, you are right that the images are of poor quality, but as there are 24 images, if we increase the quality, the file would take up too much space and we would not be able to upload the Word document.
  1. 238 Inocula, not inoculums
  2. Thank you very much, it has been changed.
  1. 244 ‘The must inoculated with Lt (Figure 2b) is smaller’ What does that mean?
  2. The sentence has been modified for better understanding.
  1. Figure 3: Sequential inoculation; not sequencial. What do the ellipses represent?
  2. The ellipses signify how there are different groups at the time of fermentation and how they evolve and change over the days.
  1. 389 Sensory results are unclear. Please re-analyse the data so as to clearly show the effect of 1) yeast treatment 2) PEF treatment.
  2. Changes have been made to the wording of this paragraph and a short conclusion has been generated for a better understanding of all the data.
  1. Figure 7: The statistical differences (letters) are not perceivable on the spider plot which is overall too crowded.
  2. Thank you for your comment, both the graph and the statistical letters have been enlarged for better understanding and visualisation.
  1. 417 It is still unclear why 5 degrees was used for the maceration. Please explain.
  2. The temperature was reduced to 5ºC to keep the must in optimal conditions and to prevent the proliferation of microorganisms during maceration, this clarification has been introduced in the article.
  1. 425 – 427. How do you explain the differences in the population of wild yeast counts between the same PEF but different yeast treatment at early stages of fermentation?
  2. The wild yeasts in the PEF-treated must were 3-log for non-Saccharomyces and 5-log for Saccharomyces. Once we had this information, we inoculated the selected yeasts with a population density of about 6-log in both the PEF-treated and untreated must, which led to a different evolution and implantation of the yeasts as well as their analytical results.

Round 2

Reviewer 2 Report

Dear authors, your manuscript has improved but my concerns regarding the scientific scrutiny and statistical analysis of the data still remain. Besides the representation and the interpretation of the microbial population data, I am concerned about the sensory analysis. With regards to the latter, the effects of the two factors (i.e. PEF and yeast treatment) are still unclear, and the drawn conclusions appear overly vague and unsupported by the data. While the univariate analysis of the sensory parameters is a common approach, it does not answer the question about the sensory effect of PEF per yeast treatment. More appropriate statistical analysis should be considered (e.g. 2-way anova). Desipte the enlarged figure, 12 treatments of the spider plot renders the figure crowded and illegible. Upon expressing my concerns, I leave it to the editor to make the final decision on this issue. 

Author Response

Thank you for your input, we have performed an ANOVA separating each variability factor: Must treatment with PEF/no must treatment and fermentative biotechnology (6 yeasts evaluated), for a better understanding of the data in a supplementary table and we have put it as an appendix at the end of the article.